# Association of Multidrug Resistance Bacteria and Clinical Outcomes of Adult Patients with Sepsis in the Intensive Care Unit

**DOI:** 10.3390/tropicalmed7110365

**Published:** 2022-11-09

**Authors:** Khalid Ahmad Al-Sunaidar, Noorizan Abd Aziz, Yahaya Hassan, Shazia Jamshed, Mahendran Sekar

**Affiliations:** 1Department of Clinical Pharmacy and Pharmacy Practice, Faculty of Pharmacy and Health Sciences, Royal College of Medicine Perak, Universiti Kuala Lumpur, Ipoh 30450, Perak, Malaysia; 2Department of Clinical Pharmacy, Faculty of Pharmacy, Universiti Teknologi MARA, Puncak Alam 42300, Selangor, Malaysia; 3Department of Clinical Pharmacy and Practice, Faculty of Pharmacy, Universiti Sultan Zainal Abidin, Besut 22200, Terengganu, Malaysia; 4Department of Pharmaceutical Chemistry, Faculty of Pharmacy and Health Sciences, Royal College of Medicine Perak, Universiti Kuala Lumpur, Ipoh 30450, Perak, Malaysia

**Keywords:** multidrug resistance organism, sepsis, adequate empirical antibiotics, source of infection, APACHE II, ICU length of stay, predictors, risk factors, mortality

## Abstract

Background: Multi-drug resistance organisms (MDRO) often cause increased morbidity, mortality, and length of stays (LOS). However, there is uncertainty whether the infection of MDRO increase the morbidity, mortality, and ICU-LOS. Objective: This study was performed to determine the prevalence of MDRO in the ICU, the site of infection, and the association of MDRO or site of infection with mortality. The secondary outcome was determined by ascertaining the association of MDRO or site of infection with ICU-LOS. Methods: A retrospective cohort study was performed with adult sepsis patients in the ICU. Univariate and multivariate (MVA) logistic regression with cox regression modeling were performed to compute the association of MDRO with ICU mortality. MVA modelling was performed for ICU-LOS predictors. Results: Out of 228 patients, the isolated MDRO was 97 (42.5%), of which 78% were Gram-negative bacteria. The mortality rate among those with MDRO was 85 (37.3%). The hospital acquired infection (HAI) was a significant predictor for ICU-LOS in univariate linear regression (R^2^ = 0.034, *p* = 0.005). In MVA linear regression, both *Enterococcus faecalis* infection and *Acinetobacter baumannii* (AC)-MDRO were predictors for ICU-LOS with (R^2^ = 0.478, *p* < 0.05). In the univariate cox regression, only the infection with AC-MDRO was a risk factor for ICU-mortality with [HR = 1.802 (95% CI: 1.2–2.706; *p* = 0.005)]. Conclusions: Identifying risk factors for MDRO addresses the appropriate administration of empirical antibiotics and allows to effectively control the source of infection, which would reduce mortality and ICU-LOS. The usage of broad-spectrum antibiotics should be limited to those with substantial risk factors for acquiring MDRO.

## 1. Introduction

Clinical studies have consistently agreed that there is an increased risk of mortality in patients with MDRO infection relative to those with non-MDRO infection. This could be related to the inappropriate use of antibiotics in the empirical stage [1]. Patients in the ICU are at increased risk of acquiring MDRO, as it seems to be more prevalent in the ICU than other wards, and thus patients are at increased risk of infection and prolonged hospital stay. This is particularly observed in patients who are immunocompromised, with organ transplantation, a history of antibiotics exposure, and with central venous catheters [2,3]. In addition, the rapid growth of MDRO had avoided the delivery of appropriate empirical antibiotics, which is the key factor of outcomes in severe patients. The increasing rate of antibiotic resistance is directly related to increased mortality, morbidity, and the cost of healthcare-associated infection, especially in the ICU [4]. Moreover, MDRO infection is known to be the main cause of inadequate empirical antibiotic therapy in the ICU [5]. In the USA, the annual incidence of antibiotic-resistant bacteria in critically ill patients is associated with more than 700,000 HAI, while in Europe, a higher incidence of *Carbapenemase*-*producing Enterobacteriaceae* (CPE) has been reported, specifically *Carbapenem-hydrolysing oxacillinase-48* (OXA-48) and *New Delhi Metallo-Betalactamase* (NDM)-*producing Enterobacteriaceae* associated infection [5,6]. *Acinetobacter baumannii* (AC), *Pseudomonas*, and *Enterobacteriaceae* MDRO are considered to be the most detrimental factors in the ICU, mostly combined with HAI or nosocomial infection [7,8,9,10,11]. 

Based on the Extended Prevalence of Infection in Intensive Care (EPIC) II study, which reported that ICU infection incidence was 51%, wherein the major source of infection was of a respiratory source (64%), the main isolated organism was *Staphylococcus aureus* (20.5%), and the Gram-negative organism constituted 62.2% (e.g., *Escherichia coli*, *Enterobacter* spp., *Klebsiella* spp., *Pseudomonas* spp., and *Acinetobacter* spp.). This is therefore pertinent in the context of a global antibiotic resistance scenario, with extensively affected regions being South-East Asia and the Middle East, where antibiotics can be easily procured over the counter and even without prescription [5,6,10,12,13,14]. 

Because of the alarming increasing trend of Gram-negative bacteria, especially MDRO *Enterobacteriaceae* with ESBL, the selection of antibiotics to target the ESBL-producing Gram negative bacteria should be based not only on the total use of antibiotics in hospitals, but also on the inappropriate use of fluoroquinolones and second or third-generation cephalosporins [4,13,14,15,16,17]. 

In one meta-analysis, patients with MDRO-*Carbapenem-resistant Enterobacteriaceae* (CRE), especially *Carbapenem-resistant Klebsiella pneumoniae* (CRKP), were reported to have a higher mortality than patients with carbapenem-sensitive bacteria [18]. Several studies have investigated the substantial association of MDRO between mortality or ICU-LOS. However, the current research, in different settings, shows that the clinical course of critically ill patients may be influenced by a few other factors post infection with MDRO, and thus have different consequences. The clinical and microbiological characteristics of ICU sepsis patients are not well known and might be different from the general population. Moreover, with a higher incidence of hospitalization and antibiotic exposure, the prevalence of MDRO over ICU sepsis patients is high [19]. Therefore, determining the causative microorganisms and their antibiotic susceptibility in this unit is important to both guide empirical treatment and to reduce mortality and morbidity. The current research analyzed the relationship of MDRO bacteria and their predictors with the risk factors or clinical outcomes, i.e., mortality and ICU-LOS. The current study was primarily performed to determine the association between MDRO and site of infection among critically ill patients with their predicators or risk factors of ICU mortality. The secondary outcome was to determine the association of MDRO or site of infection on ICU-LOS. The current data are of significance in the context of the Malaysian health care setting in order to augment the mindfulness surrounding the impact of sepsis across the country and thus to strengthen the requirement of continuous research work into prophylactic and therapeutic areas for sepsis, as well as to pave the way for resource allocation.

## 2. Materials and Methods

### 2.1. Study Design and Settings

This cohort study was performed in the ICU department of a tertiary hospital (0.526 square kilometres) in Selangor, Malaysia, with an observational retrospective design. The hospital is a major tertiary hospital located in Selangor state (130 acres) (on the west coast of Peninsular Malaysia). It consists of 620 beds and offers secondary and tertiary services for health care [20]. Before the commencement of the study, prior approval from the local ethics committees of the Research Ethics Committee (REC) and Research Management Institute (RMI), UiTM Shah Alam, was obtained. Information on patients was obtained from the ICU and pharmacy departments. Data were collected from the hospital’s computerised system/medical records of patients diagnosed with sepsis based on ICD-10 or three criteria for systemic inflammatory response syndrome (SIRS) classification and for those patients admitted between 2015 and 2016. The SIRS criteria were as follows: core temperature > 38 °C or <36 °C (>100.4 °F or <96.8 °F), elevated heart rate (>90 beats/min) (tachycardia), respiratory rate > 20 breaths/min or PaCO_2_ < 32 mm Hg or mechanical ventilation for acute respiratory process (tachypnoea), and WBC count > 12,000/mm^3^ (12 × 109 cells/L) or <4000/mm^3^ (4 × 109 cells/L) or >10% immature neutrophils [20].

The researcher confirmed the screening for the signs of sepsis, and the MDRO isolation or site of infection was counterchecked by the ICU clinician, followed by doublechecking of the data from the computer systems through the patient file records of the relevant department [12,13,14]. To determine whether the infection was HAI or CAI, the definition of Louis et al. (1995) was followed [21]. The results were screened for all of the microbiological samples collected after the patients were admitted to ICU and before they received empirical antibiotics in order to decide if the isolated microorganism was infected or colonized. A positive culture isolated from normally sterile sites, including the blood, was considered infected. The positive culture from the sputum samples was considered infected if it had three SIRS criteria. The isolates from non-sterile sites such as the urine or wounds were considered infected if accompanied with documented infection at the site of isolation. All of the other positive cultures that did not meet these criteria were considered colonized and were thus not included as active infection isolates. MDRO was identified as microorganisms not sensitive to at least one antimicrobial agent in at least three different antimicrobial categories [22]. Patients included in this study stayed in the ICU for at least three days or more, and were tracked for empirical antibiotic therapy, MDRO, site of infection, source of infection, vital signs, and clinical information with organ function parameters for up to seven days [13,14].

### 2.2. Definition of Variables, Inclusion, and Exclusion Criteria

Patients with a reported infection were identified by means of an electronic microbiology database analysis of the medical records, followed by clinician-verified results. 

The variables included demographic data, comorbidity, history of antibiotic use prior to admission to the ICU, history of surgery, time of surgery, mechanical ventilation (duration), site of infection (hospital or community-acquired infection), and source of infection (e.g., respiratory, surgical, or urinary tract infection (UTI)). It should be noted that the adequacy of empirical antibiotics evaluated based on the isolated microorganisms had to be sensitive to at least one of the combined empirical antibiotics, and that the time, dose, and frequency of administration were in line with the local/international guidelines. 

In addition, the daily records of the normal functioning and profiling function of the organs (e.g., kidney, liver, cardiac, and others, which were determined daily for each patient according to the physician clinical record sheets), laboratory findings (e.g., renal function profile, liver function profile, cardiac enzymes, and blood profiles), and (“Acute Physiology and Chronic Health Evaluation II”) APACHE II severity index assessment sheets were considered. APACHE II was calculated on the first day of admission using an online clinical calculator and included the worst vital sign records. For the isolated microorganisms, the different sources of culture (blood, sputum, wound, tissue, and urine) with sensitivity or resistant patterns for each source were studied. The same culture source could be repeated in order to obtain the isolation or different isolation patterns. 

As follows, the inclusion criteria were:Patients above 18 years of age and non-pregnant females who had been admitted to the ICU for at least 3 days to obtain antibiotic culture sensitivity results [23].ICU patients who were diagnosed with sepsis or exhibited three signs of SIRS.Patients administered with antibiotics in the ICU.Only the first admission could be included

Criteria for patient exclusion. The conditions for exclusion were as follows:More than 2 weeks of ICU hospitalisation (only the first 7 days were included in the data set for patients who had stayed for more than 7 days but less than 14 days).Incomplete data or documents that were missing; andPatients with febrile neutropenia, cystic fibrosis, burns, or HIV (absolute neutrophil count < 1000 cells/mm^3^).

### 2.3. Statistical Analysis

SPSS^®^ version 23.0 (IBM, New York, USA) for Windows^®^ was used for the data analysis, descriptive analysis (percentage and frequency), and categorical (mean ± SD for normal distributed variables) and (median and range for non-normal distributed variables) continuous data variables. The level of significance was set at *p* < 0.05. The *t*-test or Mann–Whitney u test were used to compare the ongoing data of the two classes. The ANOVA or Kruskal–Wallis test compared the data, such as demographic data, baseline clinical characteristics, comorbidities, history of antibiotics, source or site of infection, and MDRO, calculated with either the ICU-LOS or APACHE II score, of the ≥3 groups. Using χ^2^ or Fisher’s exact test, the discrete data were compared using demographic data, baseline clinical characteristics, comorbidities, antibiotic history, source or site of infection, and mortality-related MDRO. 

Univariate and multivariate logistic/Cox regressions were used to analyse the risk factors for mortality by using backward and forward methods based on goodness-of-fit principles. Using the Hosmer–Lemeshow test, the model was assessed. To establish the predictors for ICU-LOS, simple linear regression and multiple linear regression were applied. Based on the principles of best fit, backward, stepwise, and forward methods were chosen during the selection of the variables.

Practice variations inevitably and appropriately were expected when clinicians considered the needs of individual patients, the resources available, and the limitations unique to an institution or type of practice. By using post hoc stratification restriction, the contributing factors were controlled effectively or were reduced. Moreover, by using forward and backward statistical modelling for the best fit, multivariate analysis was conducted to analyse the possible effects of one variable, while simultaneously optimising for the effects of several other factors.

### 2.4. Ethical Approval and Consent to Participate

Ethical approval was obtained from the Research Ethics Committee (REC), Research Management Institute (RMI) UiTM Shah Alam [o.600-RMI (5/1/6)], and the Medical Research and Ethics Committee (MREC) and Ministry of health (MOH) via the National Medical Research Register (NMRR) No. (NMRR-14-1400-22268). The confidentiality of the patients’ data was ensured and no intervention was taken for patient management. Only the researcher had access to the online patient’s records/data anonymously based on the inclusion and exclusion criteria, no patients were involved during this study, and unnecessary informed consent was obtained from patients according to the UiTM Shah Alam Research Ethics Committee (REC) approval. All of the methods were carried out in accordance with the relevant guidelines and regulations/declaration of Helsinki.

## 3. Results

A total of 365 patients diagnosed with sepsis or who demonstrated signs of sepsis were admitted to the adult ICU ward during the study period. Only 228 patients out of 365 met the inclusion criteria. 

Demographic and clinical characteristics association with mortality is shown in Table 1, while demographic and clinical characteristics association with MDRO infection is shown in Table 2. A total of 119 (52.2 %) males and 74 (32.5 %) females were included and there were significant associations between the patient races and mortality (*p* = 0.03), while MDRO was not significant variable for ICU mortality, as shown in Table 1. There was no significant difference between MDRO and non-MDRO groups in terms of demographic data (age, gender, or race), comorbidities, type of surgery, MV, history of antibiotics used, and mortality rate, as seen in Table 2. Among the patients, 191 (83.8%) had septic shock and 83 (36.4%) had MDRO. A significant difference was observed between MDRO infection and community acquired infection (CAI) for 118 (51.8%) (*p* < 0.01) (Table 2). The mortality rate in those with CAI was 42.5%, while in those with HAI was 42.1%, as shown in Table 1. From (228) the total patients, there were 130 cases (57%) of positive isolated microorganisms. The total prevalence of MDRO was 97 (42.5%), of which Gram-negative bacteria were 78% and the rest were Gram-positive (22%). The mortality rate among those with MDRO was 85 (87.6%) (Table 1 and Table 3). The distribution of common MDRO was as follows, *Acinetobacter/MDRO.AC* 35 (15.4%), *Klebsiella pneumonia/Klebsiella* spp./*ESBL-Klebsiella*, 32 (14.0%), *P. aeruginosa* 17 (7.5%), *Enterococcus faecalis* 14 (6.1%), *Staphylococcus aureus-MRSA* 13 (5.7%), and *Enterobacteriaceae/Citrobacter koseri (diverse)-ESBL* 13 (5.7%) (Figure 1).

As shown in Table 3, the most prevalent MDRO was *Acinetobacter* spp. *(AC)* (15.35%) and the major source of isolated MDRO was from the blood (37.14%). MDRO-AC was resistant to all antibiotic (26 sample) imipenem (1 sample), polymyxin-netilmicin (2 samples), sulperazone-unasyn, piperacillin/tazobactam, amikacin, ciprofloxacin, and ceftazidime (1 sample). In addition, the isolated samples are shown in Figure 2, which illustrates the isolated blood microorganisms, of which the most isolated MRDO was AC-MDRO.

In the univariate analysis, there were significant associations between MDRO with history of surgery before ICU admission, surgery as a source of infection, skin-soft tissue infection, inadequate empirical AB, ICU-LOS, and history hospital stay before ICU (*p* < 0.05), as in Table 2.

Moreover, in the univariate analysis, all bacterial isolates were not significantly related with survival. However, there was only a significant association between the isolation of MDRO-AC bacteria and ICU-LOS (*p* < 0.001), as shown in Table 4. In addition, in simple linear regression, the model of MDRO-AC was significantly associated as a predictor for ICU-LOS with R^2^ = 0.046 and B coefficient = 5.330 (95% CI: 2.155–8.505; *p* = 0.001). A patient who acquired MDRO-AC was more likely to stay in the ICU for 5.3 days compared with the patients who did not have the same bacterial infection, as seen in Table 5. Meanwhile, in the simple linear regression, *Enterococcus faecalis* infection was a predictor for ICU-LOS with R^2^ = 0.034 and B coefficient = 6.846. The patients who acquired infection with *Enterococcus faecalis* were more likely to have increments in ICU-LOS of 6.8 days, as seen in Table 5. Furthermore, in the multivariable linear regression, both *Enterococcus faecalis* and MDRO-AC were significant predictors (R^2^ = 0.478) for increasing the ICU-LOS, B-coefficient = 4.062 (95% CI: 412–7.713; *p* = 0.029) and B-coefficient = 2.554 (95% CI: 0.064–5.044, *p* = 0.044), respectively, as shown in Table 5. On the other hand, in the univariate cox regression, only the infection with MDRO-AC was a risk factor for ICU mortality (HR = 1.802; 95% CI: 1.2–2.706; *p* = 0.005). This could explain the risk of death, which might be increased by 80% in the case of infection with MDRO-AC. Moreover, the CAI as site of infection was the ICU mortality risk factor (HR = 1.389, 95% CI 1.041–1.854, *p* = 0.026). In addition, in the multivariate cox regression, infection with only MDRO-AC increased the risk of death by 89.8% (HR = 0.102; 95% CI: 0.013–0.780; *p* = 0.028), as seen in Table 6.

## 4. Discussion

The current research has identified the clinical characteristics of infection associated with MDRO in ICU, as well as the predictors with risk factors for mortality and ICU-LOS. The majority of patients were male, elderly, and diagnosed with septic shock. The most prevalent MDRO was Gram-negative bacteria, namely MDRO-AC and *Klebsiella pneumoniae/ESBL Klebsiella pneumoniae,* which were isolated from the blood cultures. The source of MDRO infection was surgery, abdominal infection, and skin and soft tissue (SSTI) infections such as diabetic foot ulcer (DFU). In addition, this study found that inadequate empirical antibiotics were associated with MDRO infection. It appears that patients who received inadequate empirical antibiotics were more likely to develop MDRO infection. Furthermore, the history of hospital stays before ICU admission increased the vulnerability of patients to acquire MDRO infection. Furthermore, MDRO infection was more observed among those with a longer ICU stay than those with a shorter period. This explains that the length of hospital or ICU stay demonstrated an increased risk of MDRO infection. The current finding is consistent with the study done in China, which evaluated the risk factors for mortality in ICU patients with *Acinetobacter *baumannii** VAP [23].

The current research also reported that AC- MDRO infection was a risk factor for ICU mortality. Similarly, other related retrospective studies were conducted to analyze the risk factors for mortality in ICU sepsis patients with *Acinetobacter baumannii*-VAP. It has been shown that Gram-negative bacteria were the most common pathogens (46.0%) and were associated with increased ICU mortality [22]. Furthermore, the coherent findings of the Chernen et al. (2013) study reported that there was an increase in Gram-negative infection from 38.26% to 48.1%. The share of *Klebsiella pneumoniae* isolates and *Acinetobacter* spp. were amplified from 8.1 to 18.9% [24]. Likewise, Li et al. (2017) conducted a surveillance study of nosocomial infection in the intensive care units of 177 hospitals. The isolation rates of Gram-negative bacteria MDRO were as follows *Carbapenem-resistant Acinetobacter baumannii* 80.53%, *Carbapenem-resistant Pseudomonas aeruginosa* 39.94%, *Carbapenem-resistant Klebsiella pneumoniae* 24.86%, and *Carbapenem-resistant Escherichia coli* 9.23% [25]. The current study by MVA found that both *enterococcus faecalis* and AC-MRDO infection were a significant independent predictor for ICU-LOS (*p* = 0.005). The similar findings of another study conducted to measure the clinical outcomes of *Enterococcus faecalis* reported that prevalence was 57.6% in ICU and the mortality was significantly associated with polymicrobial bacteria and ICU-LOS [26]. Meanwhile, another prospective, observational, multicentre study informed that the isolated *Enterococcus* spp. in ICU was 10.2% and the predominant species was *Enterococcus faecalis* (82.4%) [27].

Moreover, the recent study stated the infection of *Enterococcus faecalis* was a risk factor for ICU mortality. A similar study compared the clinical outcome differences between *Vancomycin-resistant Enterococcus* caused by *Enterococcus faecalis or Enterococcus faecium. Enterococcus faecium* was more resistant to antibiotics (ampicillin and teicoplanin) and showed a higher mortality [27]. Meanwhile, the existing study found that there was no significant association between multi-drug resistance microorganisms (MDRO) and predictors for survival. Similarly, a retrospective observational cohort study was conducted by Lye et al. (2012) with MDRO Gram-negative bacteria in severe sepsis and septic shock patients at two large Singaporean hospitals. The study found, through multivariable analysis, that MDRO was not associated with mortality, but rather related to longer stays of 6.1 days in hospital for LOS in survivors [28]. Furthermore, the consistent findings by a prospective, observational study conducted in sepsis in the ICU to measure antibiotic bacterial resistance have shown that patients with MDRO significantly received inadequate empirical antibiotics more frequently and long ICU-LOS than patients with sepsis due to non-MDRO with higher mortality (*p* < 0.05) [29]. In addition, the current findings are in accordance with a prospective study that reported that patients with MDRO infection would have a higher chance to receive inadequate empirical antibiotics [30]. In addition, the findings of other retrospective studies are consistent with the current study, which were conducted to measure the clinical outcomes of nosocomial Gram-negative bacteria in ICU. The results showed that exposure to carbapenem increased the hazard risk (HR = 4.087) of acquiring the infection of *Carbapenem resistant AC*- MDRO [30,31,32].

## 5. Conclusions

The outcomes of the current research indicated that recognizing the risk factors for MDRO infection could lead to more effective use of empirical antibiotics, thereby minimizing the source of infection, thus lowering mortality and ICU-LOS. The high prevalence of MDRO organisms has a role in the patients’ mortality. The infection of MDRO is also related to poor clinical outcomes and longer ICU-LOS. Furthermore, inadequate empirical antibiotic therapy was a major contributor to MDRO infection. The predominant microorganisms were Gram-negative bacteria with MDRO organisms, e.g., AC-MDRO. The overuse of broad-spectrum antibiotics should be limited to those with significant risk factors for acquiring MDRO organisms. This addresses the significance of antimicrobial stewardship programs. Antibiotics guidelines are expected to be in concordance with an infection control strategy, thereby the emergence and transmission of MDRO infection is minimized. The local and regional guidelines must be in line with the local epidemiological and microbiological data. Future recommendations must envisage the analysis of the available regulations and guidelines for improving the management of MDRO infection in critically ill sepsis patients.

## 6. Limitations

The small sample size in our study may have reduced the possibility to show a difference in mortality between the two groups. A larger sampling size may be needed to show the difference. Because of the retrospective design of this study, data may be incomplete or missed during the retrieval of information. This study was based on one ICU in a single tertiary hospital for a limited period and therefore may not be fully representative of other hospitals. The criteria of sepsis definition were not in agreement with the latest definition by (Sepsis-3) [33].

## Figures and Tables

**Figure 1 tropicalmed-07-00365-f001:**
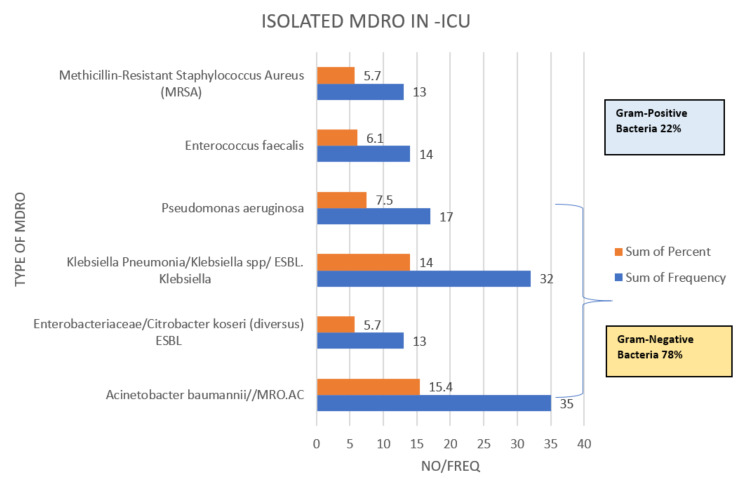
Most common isolated Multi drug resistant organisms (MDRO) in Sepsis ICU patients.

**Figure 2 tropicalmed-07-00365-f002:**
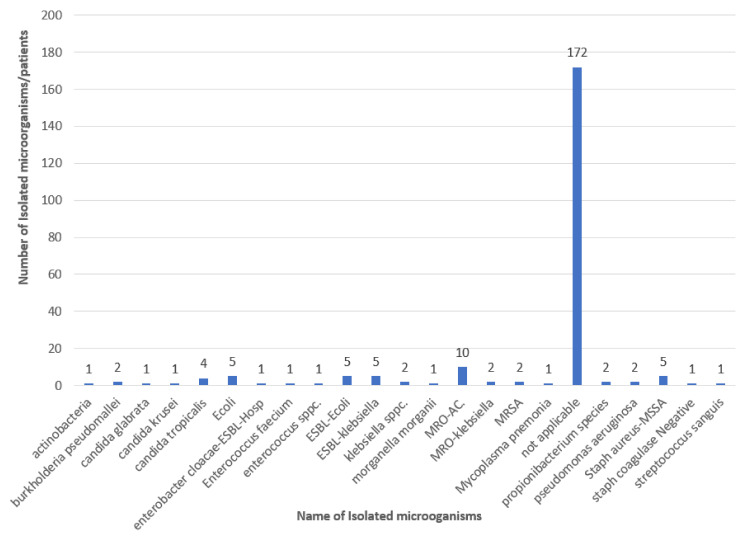
The total isolated microorganisms from blood culture. Not applicable = Number of samples without isolated microorganisms from the whole patients who were admitted to ICU (228).

**Table 1 tropicalmed-07-00365-t001:** The Association of Demographic and Baseline Clinical Characteristics of Entire Sample with Mortality.

Parameters/ Outcomes	Non-Survivor (*n* = 193) No (%)	Survivor (*n* = 35) No (%)	*p* Value
Age			
<48	52 (22.8)	5 (2.2)	0.405 a *
48–58	44 (19.3)	11 (4.8)	
59–66	47 (20.6)	9 (3.9)	
67+	50 (21.9)	10 (4.4)	
Total	193 (84.6)	35 (15.4)	
Gender			
Male	119 (52.2)	18 (7.9)	0.266 b *
Female	74 (32.5)	17 (7.5)	
Total	193 (84.6)	35 (15.4)	
Race			
Malay	122 (53.5)	19 (8.3)	0.030 # a *
Chinese	20 (8.8)	10 (4.4)	
Indians	39 (17.1)	5 (2.2)	
Others	12 (5.3)	1 (0.4)	
Total	193 (84.6)	35 (15.4)	
Surgery			
Yes	127 (55.7)	25 (11)	0.565 b *
No	66 (28.9)	10 (4.4)	
Total	193 (84.6)	35 (15.4)	
Hx. Time of surgery			*
Current–1 week	107 (69.5)	21 (13.6)	0.606 a *
>1 week–6 month	13 (8.4)	2 (1.3)	
>6 month	8 (5.2)	3 (1.9)	
Total	128 (83.1)	26 (16.9)	
Type of surgery			
Skin soft T.S /DFU/	27 (17.3)	5 (3.2)	0.690 a *
Orthopaedics/Polytrauma	15 (9.6)	3 (1.9)	
Neurosurgery	45 (28.8)	7 (4.5)	
Abd surgery/liver & Biliary sepsis/	37 (23.7)	8 (5.1)	
Others(cardiac-Urological-tracheostomy)	6 (3.8)	3 (1.9)	
Total	130 (83.3)	26 (16.7)	
Classification of Infection Site			
Community acquired infection	97 (42.5)	21 (9.2)	0.359 a *
Healthcare associated infection	96 (42.1)	14 (6.1)	
Mental state			
Alert	21 (9.2)	7 (3.1)	0.295 a *
Confused	171 (75.0)	28 (12.3)	
Coma	1 (0.4)	0	
Total	193 (84.6)	35 (15.4)	
GCS-Day1			
(Severe GCS)	169 (74.1)	28 (12.3)	0.256 a *
(Moderate GCS)	8 (3.5)	1 (0.4)	
(Mild GCS)	16 (7.0)	6 (2.6)	
Total	193 (84.6)	35 (15.4)	
MDRO	Yes 85 (37.3) No 108 (47.4)	12 (5.3) 23 (10.1)	0.354 a *

*—type of test, a *—chi square, b *—fisher exact, # significant value (*p* < 0.05).

**Table 2 tropicalmed-07-00365-t002:** Univariate association of baseline clinical characteristics and MDRO.

Characteristics	Total *n* = 228 No (%)	MDROs *n =* 97 No (%)	Non-MDROs *n* = 131 No (%)	*p* Value *
Age				
<48	57 (25)	23 (10.1)	34 (14.9)	0.797 * a
48–58	55 (24.1)	23 (10.1)	32 (14.0)	
59–66	56 (24.6)	27 (11.8)	29 (12.7)	
67+	60 (26.3)	24 (10.5)	36 (15.8)	
Male Gender	137 (60.1)	59 (25.9)	78 (34.2)	0.892 * b
Race				
Malay	141 (61.8)	61 (26.8)	80 (35.1)	0.531 * a
Chinese	30 (13.2)	14 (6.1)	16 (7.0)	
Indians	44 (19.3)	15 (6.6)	29 (12.7)	
Others	13 (5.7)	7 (3.1)	6 (2.6)	
Hx. Of Surgery during ICU admission	152 (66.7)	74 (32.5)	78 (34.2)	0.01 # * a
Type of Surgery				
Skin & soft tissue infection	17 (7.5)	9 (3.9)	8 (3.5)	0.075 * b
Orthopaedics	6 (2.6)	3 (1.3)	3 (1.3)	
Neurosurgery	52 (22.8)	19 (8.3)	33 (14.5)	
Abdominal	36 (15.8)	25 (11)	11 (4.8)	
Cardio	3 (1.3)	1 (0.4)	2 (0.9)	
DFU-Amputation-gangrene	15 (6.6)	8 (3.5)	7 (3.1)	
Biliary sepsis	2 (0.9)	1 (0.4)	1 (0.4)	
Polytrauma-trauma	12 (5.3)	5 (2.2)	7 (3.1)	
Urological, genital	4 (1.8)	2 (0.9)	2 (0.9)	
UGIB ¹	3 (1.3)	1 (0.4)	2 (0.9)	
Tracheostomy—others	2 (0.9)	0 (0)	2 (0.9)	
Comorbidities				
DM	111 (48.7)	45 (19.7)	66 (28.9)	0.593 * a
HTN	152 (66.7)	61 (26.8)	91 (39.9)	0.322 * a
Asthma	16 (7.0)	5 (2.2)	11 (4.8)	0.436 * b
COPD	8 (3.5)	5 (2.2)	3 (1.3)	0.290 * b
CAD	48 (21.1)	18 (7.9)	30 (13.2)	0.512 * a
CHF	34 (14.9)	10 (4.4)	24 (10.5)	0.132 * b
CRF	36 (15.8)	15 (6.6)	21 (9.2)	1.000 * b
Co-Malignancy	8 (3.5)	5 (2.2)	5 (1.3)	0.290 * b
Liver disease	21 (9.2)	11 (4.8)	10 (4.4)	0.362 * b
GCS ^2^				
Severe	197 (86.4)	80 (36.4)	114 (50)	0.950 * a
Moderate	9 (3.9)	4 (1.8)	5 (2.2)	
Mild	22 (9.6)	10 (4.4)	12 (5.3)	
Mental Status				
Alert	28 (12.3)	13 (5.7)	15 (6.6)	0.454 * a
Confused	199 (87.3)	83 (36.4)	116 (50.9)	
Coma	1 (0.4)	1 (0.4)	0 (0)	
Hx of AB used during last two weeks before ICU admission	174 (76.3)	75 (32.9)	99 (43.4)	0.875 * a
Received MV	226 (99.1)	97 (42.5)	129 (56.6)	0.509 * a
Diagnosis				
Sepsis	34 (14.9)	11 (4.8)	23 (10.1)	0.062 * b
Severe Sepsis	3 (1.3)	3 (1.3)	0 (0)	
Septic shock	191 (83.8)	83 (36.4)	108 (47.4)	
Site transferred to ICU				
ED ^3^	78 (34.2)	27 (11.8)	51 (22.4)	0.372 * a
MW ^4^	81 (35.5)	37 (16.2)	44 (19.3)	
SW ^5^	65 (28.5)	31 (13.6)	34 (14.9)	
Others	4 (1.8)	2 (0.9)	2 (0.9)	
Classification of Infection Site				
Community acquired infection	118 (51.8)	40 (17.5)	78 (34.2)	0.007 # * a
Healthcare associated infection	110 (48.2)	57 (25.0)	53 (23.2)	
Source of Infection				
RTI	131 (57.5)	53 (23.2)	78 (34.2)	0.499 * a
UTI	29 (12.7	15 (6.6)	14 (6.1)	0.318 * b
ABD	60 (26.3)	30 (13.2)	30 (13.2)	0.223 * a
Skin soft T.S inf. (SSTIs)	50 (21.9)	28 (12.3)	22 (9.6)	0.035 # * a
Surgery	124 (54.4)	62 (27.2)	62 (27.2)	0.016 # * a
Unknown	14 (6.1)	7 (3.1)	7 (3.1)	0.587 * b
Adequate empirical AB	64 (28.1)	14 (6.1)	21.9)	<0.001 # * a
ICU death	193 (84.6)	85 (37.3)	108 (47.4)	0.354 * a
ICU-LOS (day)				
<5.0	56 (24.6)	16 (7.0)	40 (17.5)	0.004 # * a
5.0–6.0	48 (21.1)	18 (7.9)	30 (13.2)	
7.0–11.4	67 (29.4)	28 (12.3)	39 (17.1)	
11.5+	57 (25.0)	35 (15.4)	22 (9.6)	
Hosp-LOS before ICU (day)				
Zero	99 (43.4)	36 (15.8)	63 (27.6)	0.010 # * a
1–2	72 (31.6)	27 (11.8)	45 (19.7)	
+3	57 (25.0)	34 (14.9)	23 (10.1)	
MV ^6^ Duration (day)				
1–3	39 (17.3)	16 (7.1)	23 (10.2)	0.221 * b
4–6	71 (31.4)	24 (10.6)	47 (20.8)	
7	88 (38.9)	44 (19.5)	44 (19.5)	
>7	28 (12.4)	13 (5.8)	15 (6.6)	

¹ Upper gastrointestinal bleeding, ^2^ Glasgow coma scale, ^3^ Emergency department, ^4^ Medical ward, ^5^ Surgical ward, ^6^ Mechanical ventilation, *—type of test, * a—chi square, * b—fisher exact, # significant value (*p* < 0.05).

**Table 3 tropicalmed-07-00365-t003:** The sensitivity patterns of MDRO organisms and their main sources of isolated culture samples.

MDRO-Organisms	Total Isolation from 228 Patients (N, %)	Source of Isolated Culture Sample (N, %)	Sensitivity to Antibiotics	Resistant to Antibiotic
MRSA *a	13 (5.7%)	Blood = 4 (30.76) Nasal = 2 (15.38) Sputum = 7 (53.84)	GEN = 1 IPM = 1 LEZ = 2 MUP = 3 OXA = 1 VAN = 5	ALL * = 2 OXA = 9
*Pseudomonas aeruginosa*	17 (7.45%)	Blood = 4 (23.52) Knee aspiration = 1 (5.88) Sputum = 10 (58.82) CSF = 2 (11.76)	AMK = 1 AMK- TAZ -CIP-CFP-CAZ = 1 AMK-CIP-CXM-GENT-SXT = 1 CEP = 5 IPM = 4 SPZ-TAZ-AMK-CIP-GEN-CAZ = 3	AMK-TAZ-CIP- CEP-CAZ = 1 AMC = 5 AMK-CIP-CXM-GENT = 1 GENT = 1
*Klebsiella pneumoniae*/ESBL *Klebsiella pneumoniae*	32 (14%)	Blood = 18 (56.25) Wound = 3 (9.37) Sputum = 9 (28.12) Tissue-CSF = 1 (3.12) Urine = 1 (3.12)	AMK-TAZ-CIP-FEB-CAZ = 1 AMK-CIP-CXM-GEN = 1 CRE-CIP = 1 CAZ-TAZ-IPM = 2 IPM = 8 IPM-MEM-ETP-AMK = 5 PB1 = 2 SPZ = 2 SPZ-AMC-CXM-GEN = 5	ALL * = 2 AMK-CIP-CXM-GEN-SXT = 1 AMP = 2 AMC = 3 CFP-CIP = 1 CXM-NET-AMC-AMP = 1 IPM = 1 SPZ-TAZ-AMK-CIP-GEN-CAZ = 5 SPZ-AMC-CXM-GEN = 1
*Enterococcus faecalis*	14 (6.1%)	Blood = 4 (28.57) Wound = 3 (21.42) Tissue-CSF = 3 (21.42) Urine = 4 (28.57)	AMK-CIP-CXM-GEN-SXT = 1 AMP-GEN-VAN-TGC = 6 AMP = 3 CXM-NFN-AMC-AMP = 1 VAN = 3	AMP-GEN-VAN-TGC = 1 AMP = 1 CIP = 1 GEN = 3 SPZ-TAZ-AMK-CIP-GEN-CAZ = 1 VAN = 1
MDRO-AC. *b	35 (15.35%)	Blood = 13 (37.14) Wound = 2 (5.71) Tissue-CSF = 4 (11.42) Urine = 2 (5.71) Sputum = 14 (40)	AMK = 1 IPM-MEM-ETP-AMK = 1 PB1 = 15 SPZ = 1 SPZ-TAZ-AMK = 1 TGC = 9	ALL * = 26 IPM = 1 PB1-NET = 2 SPZ-TAZ-AMK-CIP-GEN-CAZ = 1
*Enterobacteriaceae-ESBL* *c—*Escherichia coli*	13 (5.7%)	Blood = 7 (53.84) Wound = 3 (23.07) Tissue-CSF = 1 (7.69) Urine = 1 (7.69) Sputum = 1 (7.69)	AMK = 1 AMK-TAZ-CIP-CFP-CAZ = 1 CP-CIP = 1 XCM-NET-AMC-AMP = 1 IPM = 2 IPM-MEM-ETP-AMK = 3 PB1 = 1SPZ = 1 SPZ-AMC-CXM-GEN = 3	ALL * = 2 AMP = 3 CXM = 1 CXMSPZ-CIP-SXT = 2 GEN-SXT = 1 SPZ-AMC-CXM-GEN = 1

*a: MRSA: *Methicillin-resistant Staphylococcus* antibiotics, *b: *Acinetobacter baumannii* -multi resistant organisms, *c: *Extended-spectrum beta-lactamases*, ALL * = Pandrug resistant bacteria to all antibiotics, AMP = ampicillin, AMC = amoxicillin/clavulanate, AMK = amikacin, CIP = ciprofloxacin, CEP = cefepime, CAZ = ceftazidime, CXM = cefuroxime, CTX = cefotaxime, CP = carbapenem, ETP = ertapenem, GEN = gentamicin, IPM = imipenem, LEZ = linezolid, MUP = mupirocin, MEM = meropenem, NFN = nitrofurantoin, NET = netilmicin, OXA = oxacillin, SPZ = sulperazone-unasyn, TAZ = piperacillin/tazobactam, TGC = tigecycline, SXT = trimethoprim-sulphamethoxazole, PB1 = polymyxin, VAN = vancomycin.

**Table 4 tropicalmed-07-00365-t004:** Association of isolated MDRO organisms with their outcomes (mortality, APACHE II score, ICU-LOS).

Parameters-EST/Outcomes	ICU-Death	*p* Value	APATCHE II (Severity Index)	*p* Value	ICU-LOS-DAY	*p* Value
*Staphylococcus aureus* (MRSA)	Yes 12 = 5.3% No 181 = 79.4% Total 193 = 84.6%	0.697 b *	31.00 (IQR = 23.00–35.00)	0.360 f *	7.00 (IQR = 5.00–11.75)	0.196 f *
*Pseudomonas aeruginos* *a*	Yes 16 = 7.0% No 177 = 77.6% Total 93 = 84.6%	0.482 b *	31.00 (IQR = 23.00–35.00)	0.804 f *	7.00 (IQR = 5.00–11.75)	0.520 f *
*Klebsiella pneumonia* /*Klebsiella Spps*/ESBL *Klebsiella*	Yes 26 = 11.4% No 167 = 73.2% Total 193 = 84.6%	0.597 b *	31.00 (IQR = 23.00–35.00)	0.367 f *	7.00 (IQR = 5.00–11.75)	0.050 f *
Acinetobacter/MDRO.AC	Yes 29 = 12.7% No 164 = 71.9% Total 193 = 84.6%	0.799 b *	31.00 (IQR = 23.00–35.00)	0.884 f *	7.00 (IQR = 5.00–11.75)	<0.001 f *
Enterobacteriaceae/*Citrobacter koseri* (diversus) ESBL	Yes = 11 = 4.8% No = 182 = 79.8% Total = 193 = 84.6%	1.000 b *	31.00 (IQR = 23.00–35.00)	0.359 f *	7.00 (IQR = 5.00–11.75)	0.884 f *

*—chi square, b *—fisher exact test, f *—Mann Whitney u test.

**Table 5 tropicalmed-07-00365-t005:** The univariate and multivariate linear regression of MRDO organisms as predictors for increasing (ICU-LOS).

Variable	B-Coefficient	Simple Linear Regression R^2^ (95% CI)	*p* Value	Multivariable Linear Regression R^2^ (95% CI)	B-Coefficient	*p* Value
AC- MDRO * bacteria	5.330	0.046 (2.155–8.505)	0.001	0.478 (0.064–5.044)	2.554	0.044
Enterococcus faecalis	6.846	0.034 (2.049–11.644)	0.005	0.478 (0.412–7.713)	4.062	0.029
HAI Infection	3.310	0.034 (−5.61–−1.006)	0.005	–	–	–

* *Acinetobacter baumannii*—multi drug resistant organisms.

**Table 6 tropicalmed-07-00365-t006:** The univariate and multivariate cox regression risk factor for ICU mortality.

Variable	B-Coefficient	Univariate Cox Regression HR (95% CI)	*p* Value	Multivariate Cox Regression HR (95% CI)	B-Coefficient	*p* Value
AC- MDRO * bacteria	0.589	1.802 (1.2–2.7)	0.005	0.102 (0.013–0.780)	−2.278	0.028
Enterococcus faecalis	0.385	1.47 (0.831–2.6)	0.186	–	–	–
CAI Infection	0.329	1.389 (1.041–1.854)	0.026	–	–	–

* *Acinetobacter baumannii*—multi resistant organisms.

## Data Availability

The datasets used and/or analyzed during the current study are available from the corresponding author upon reasonable request. Please contact the author for data requests.

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
