# Peer review of "Association of Multidrug Resistance Bacteria and Clinical Outcomes of Adult Patients with Sepsis in the Intensive Care Unit"

_tropicalmed, 2022, doi:10.3390/tropicalmed7110365_

Round 1
Reviewer 1 Report
Line 50 "rapid existence" the meaning is not clear
Line 202 "Only 228 patients met ... criteria" Define better the number of patients included in the study.
Line 266 "was more likely to stay in the ICU with 5.3 days" The meaning is not clear
Line 279 " while in multivariate cox regression, the only infection with MDRO-AC was a protective factor" This statement is contradicts the previous statement in line 274 "On the other hand... " Please define it better
Author Response
- Line 50 "rapid existence" the meaning is not clear -Thanks for your comments, it means the rapid growth of MDRO.
- Line 202 "Only 228 patients met ... criteria" Define better the number of patients included in the study. Thanks, already improved The actual number targeted in the study was 365 patients after we screened the patient based on our inclusion and exclusion criteria, only we have 228 patients were included in the study.
- Line 266 "was more likely to stay in the ICU with 5.3 days" The meaning is not clear: Thanks for your comments. This result is based on simple linear regression as a predictor for the ICU-LOS as a statistical approach to estimate the effect of ID variables on the outcome. In this regards the acquiring of AC.MRDO will increase the ICU-LOS by 5.3 days.
- Line 279 " while in multivariate cox regression, the only infection with MDRO-AC was a protective factor" This statement contradicts the previous statement in line 274 "On the other hand... " Please define it better. Thanks for your comment. It's already improved . however, There was NO contradiction because we do two levels of cox regression, one is the univariate as in line 274 and the other multivariate cox regression as in line 279. The multivariate cox regression is used to adjust the effects of other potential factors, there have been selected all significant predictors in simple logistic regression (P=<0.05). Also, to reduce /eliminate the effects of confounders factors.

Reviewer 2 Report
In this paper by Al-Sunaidar et al, a recent retrospective clinical study is described of the incident of MDR bacterial infections in a single hospital and its impact on mortality and patient stay. The findings show an association of MDR infections with mortality, and a specific association with E. faecalis infection. While these findings are a valuable addition to the many existing antibiotic resistance surveillance studies, some points must be addressed before this is suitable for publication.
Major Points
-The tables within the paper are overall unclear and poorly organized. For example, in table 2 it is unclear how many samples are included in the 'sensitivity to antibiotics' and 'resistant to antibiotics' columns. These would be much better organized with each individual antibiotic and the number resistant to each among all isolates and within each organism
-There is no data on the antibiotics used, only that empirical antibiotic treatment was used before testing. There should be data included on which antibiotics were used for each isolate
Minor points
-How was antibiotic resistance determined? It is important to know what type of testing was used for each antibiotic. Some tables use 'ALL' for resistance to all antibiotics, but there is no list of which antibiotics were tested
-Various grammatical and vocabulary errors throughout, this manuscript needs significant English language editing
Author Response
We would like to express our heartfelt gratitude to the reviewers for taking the time to read the manuscript and provide us with constructive feedback. All of your suggestions have been taken into consideration in order to improve the quality of our manuscript.
- Table 2 it is unclear how many samples are included in the 'sensitivity to antibiotics' and 'resistant to antibiotics' columns. These would be much better organized with each individual antibiotic and the number resistant to each among all isolates and within each organism. Thanks for your comments , the table 2 and figure 2 &3 have shown the source of sample and the main antibiotic sensitivity or resistance patterns for each MDRO. When we have collected the data for each MDRO organism. After we have analyzed the data we have presented the result for each MDRO as a whole pattern of either sensitivity or resistance not as an individual sample because we are dealing with big data for more than 230 patients, it will be too lengthy paper with more than 10 figures to give individual results for each sample. if you can kindly refer to figures no 1 and 2 it will show you the number of isolated MO including the MDRO. Here I have additional figures for your reference that show the antibiotic culture sensitivity and resistance.
- There is no data on the antibiotics used, only that empirical antibiotic treatment was used before testing. There should be data included on which antibiotics were used for each isolate. Thanks for your comment. In ICU. ward when we have screened the patients' medication sheets for the empirical antibiotic used, we found they have used more than 120 different combinations as empirical antibiotic treatment. Therefore the researchers will identify the adequacy of empirical antibiotics considering the isolated MO. and coverage of empirical AB. based on culture sensitivity test. Please find here the additional table (which has been published previously in different papers) that shows the most used first and second-line empirical antibiotics.
-
Minor points
-How was antibiotic resistance determined? It is important to know what type of testing was used for each antibiotic. Thanks ( it was mentioned the type of test and source for each sample). Also, the data of this study was retrospectively collected, so the researcher only trace the medical sheets and those specifically for lab culture results that clearly mentioned whether such MO was sensitive or resistant to AB. . Some tables use 'ALL' for resistance to all antibiotics, but there is no list of which antibiotics were tested. Thanks, ALL refer in such table/ figures to all other AB which is listed in table 2 or used as empirical antibiotics
-Various grammatical and vocabulary errors throughout, this manuscript needs significant English language editing. Thanks for your comment and will submit this paper to the authorized paper language editing expert for cross-checking.

Reviewer 3 Report
Dear authors,
Your paper looks good and dedicated to very important issue. Given the fact that a personalized approach in microbiological diagnostics is impossible to apply, primarily due to the duration of the analysis, any evaluation of the results of the causative agent, antimicrobial susceptibility testing and monitoring of patients is significant.
However, it must be improved to be for publication.
Comments:
-Paper must be shortened (text of description of institutions, ethical approved...
-In my opinion, since the many tables and results are presented in article, you can omit some tables (for example Table 1) and all analyzed data only mention in text. There are no significant findings.
-All name of genus and species of bacteria must be written as: the genus is written with a capital letter and the species with a small letter, if only the genus is mentioned, it must be followed by the abbreviation spp. (for example Acinetobacter baumannii no acinetobacter baumannii ; Pseudomonas spp. no pseudomonas…)
-In the table 2, where you present the number of sensitive and resistant species you have to list them as percentages, not numbers .
-You must better explain the finding of the univariate and multivariate cox regression risk factor for ICU mortality, since that your comment was that the only infection with MDRO-AC was a protective factor???
-You have to check all abbreviations, some was not make understand
-It will be better if you can change the term site of infection with where it was acquired, site of infection is often use for localization of infection. Additionally, surgery can not be the source of MDRO infection
-If you have to explain results in Tabels it has to be in order, if you explain first table 3, not table 2, then you change the numbers of tables, all abbreviations in tables must be explain in legends
- My opinion is that patients could not met the inclusion and exclusion criteria, only one criteria (Line 201)
-Finally, it will be useful in the end of discussion to mention the new “syndrome approach” testing by molecular analyses which can give us results in few hours where it can be applied.
Author Response
We would like to express our heartfelt gratitude to the reviewers for taking the time to read the manuscript and provide us with constructive feedback. All of your suggestions have been taken into consideration in order to improve the quality of our manuscript.
- The paper must be shortened (text of the description of institutions, ethical approved. Thanks for your comments. both sections, I think, are quite reasonable.
- In my opinion, since the many tables and results are presented in article, you can omit some tables (for example Table 1) and all analyzed data only mention in the text. There are no significant findings. Thanks, for table -1 shows the patient distribution and association with mortality, while there were some significant associations in variables. Not all table results have been explained through the text. Therefore, it still can give the reader the impression of this kind of patient cohort.
- All name of genus and species of bacteria must be written as: the genus is written with a capital letter and the species with a small letter, if only the genus is mentioned, it must be followed by the abbreviation spp. (for example Acinetobacter baumannii no acinetobacter baumannii ; Pseudomonas spp. no pseudomonas…), Thanks, has been improved accordingly .
- -In the table 2, where you present the number of sensitive and resistant species you have to list them as percentages, not numbers, Thanks , they were presented by both numbers and % for much good understanding.
- You must better explain the finding of the univariate and multivariate cox regression risk factor for ICU mortality, since that your comment was that the only infection with MDRO-AC was a protective factor???, thanks , they were fully explained and the researchers have used two levels of cox regression as univariate and Multivariable Cox Regression Analysis(MVA) for ICU Risk of Mortality. (MVA) is used to eliminate any confounding factors and used to adjust the effects of other potential factors, therefore during the second run of MVA , only MDRO-AC was a protective factor with the hazard rates is less than 1 (HR=0 .102; 95% CI: .013-.780; P=.028) as in table 6.
- -You have to check all abbreviations, some were not make understood, thanks, have been improved accordingly
- It will be better if you can change the term site of infection with where it was acquired, site of infection is often use for localization of infection. Additionally, surgery can not be the source of MDRO infection, Thanks for your comments, in our manuscript we have used the site of infection to differentiate either as CAI /HAI, while the source was used to identify the source of infection as mentioned in details in Methodology part , In Our study sample of some patients with sepsis in ICU who were admitted to ICU post-surgery were susceptible to acquire MDRO, which other similar studies have reported the same
- -If you have to explain results in Tabels it has to be in order, if you explain first table 3, not table 2, then you change the numbers of tables, all abbreviations in tables must be explain in legends, Thanks , improved accordingly
- My opinion is that patients could not met the inclusion and exclusion criteria, only one criteria (Line 201), , thanks, improved accordingly
- Finally, it will be useful in the end of discussion to mention the new “syndrome approach” testing by molecular analyses which can give us results in few hours where it can be applied. thanks, no comment

Round 2
Reviewer 2 Report
Thanks for addressing previous points, no further comments.
Author Response
Thanks for your comments and assistance, all the previous comments have been addressed and improved accordingly. Meanwhile, the proofreading expert has corrected the manuscript wherever it's needed and highlighted it in blue color.

Reviewer 3 Report
Dear authors,
Some suggestions more
-Again, you must better explain the finding of the univariate and multivariate cox regression risk factor for ICU mortality, since that your comment was that the only infection with MDRO-AC was a protective factor, you must change it
In Table 3 in legend MRSA is not Methicillin-resistant sStaphylococcus antibiotics it is Methicillin-resistant Staphylococcus aureus
Author Response
Thanks for your comments and assistance, all the previous comments have been addressed and improved accordingly. Meanwhile, the proofreading expert has corrected the manuscript wherever it's needed and highlighted it in blue color.
Reviewer Comments
1-Again, you must better explain the finding of the univariate and multivariate cox regression risk factor for ICU mortality, since that your comment was that the only infection with MDRO-AC was a protective factor, you must change it,-Thanks for your comment and it has improved accordingly.
2- In Table 3 in legend MRSA is not Methicillin-resistant Staphylococcus antibiotics it is Methicillin-resistant Staphylococcus aureus, Thanks for your comment and it has improved accordingly
